# Transplacental Therapeutic Drug Monitoring in Pregnant Women with Fetal Tachyarrhythmia Using HPLC-MS/MS

**DOI:** 10.3390/ijms24031848

**Published:** 2023-01-17

**Authors:** Natalia Starodubtseva, Svetlana Kindysheva, Alyona Potapova, Evgenii Kukaev, Zulfiya Khodzhaeva, Ekaterina Bockeria, Vitaliy Chagovets, Vladimir Frankevich, Gennady Sukhikh

**Affiliations:** 1National Medical Research Center for Obstetrics Gynecology and Perinatology Named after Academician V.I., Kulakov of the Ministry of Healthcare of Russian Federation, 117997 Moscow, Russia; 2Moscow Institute of Physics and Technology, 141700 Moscow, Russia; 3V.L. Talrose Institute for Energy Problems of Chemical Physics, Russia Academy of Sciences, 119991 Moscow, Russia; 4Laboratory of Translational Medicine, Siberian State Medical University, 634050 Tomsk, Russia

**Keywords:** fetal tachyarrhythmia, therapeutic drug monitoring, digoxin, sotalol, mass spectrometry, high performance liquid chromatography with mass spectrometry

## Abstract

Fetal arrhythmia develops in 0.1–5% of pregnancies and may cause fetal heart failure and fetal hydrops, thus increasing fetal, neonatal, and infant mortality. The timely initiation of transplacental antiarrhythmic therapy (ART) promotes the conversion of fetal tachycardia to sinus rhythm and the regression of the concomitant non-immune fetal hydrops. The optimal treatment regimen search for the fetus with tachyarrhythmia is still of high value. Polymorphisms of these genes determines the individual features of the drug pharmacokinetics. The aim of this study was to study the pharmacokinetics of transplacental anti-arrhythmic drugs in the fetal therapy of arrhythmias using HPLC-MS/MS, as well as to assess the effect of the multidrug-resistance gene *ABCB1* 3435C > T polymorphism on the efficacy and maternal/fetal complications of digoxin treatment. The predisposition to a decrease in the bioavailability of the digoxin in patients with a homozygous variant of the CC polymorphism showed a probable association with the development of ART side effects. A pronounced decrease in heart rate in women with the 3435TT allele of the *ABCB1* gene was found. The homozygous TT variant in the fetus showed a probable association with an earlier response to ART and rhythm disruptions on the digoxin dosage reduction. high-performance liquid chromatography with tandem mass spectrometry (HPLC-MS/MS) methods for digoxin and sotalol therapeutic drug monitoring in blood plasma, amniotic fluid, and urine were developed. The digoxin and sotalol concentrations were determined in the plasma blood, urine, and amniotic fluid of 30 pregnant women at four time points (from the beginning of the transplacental antiarrhythmic therapy to delivery) and the plasma cord blood of 30 newborns. A high degree of correlation between the level of digoxin and sotalol in maternal and cord blood was found. The ratio of digoxin and sotalol in cord blood to maternal blood was 0.35 (0.27 and 0.46) and 1.0 (0.97 and 1.07), accordingly. The digoxin concentration in the blood of the fetus at the moment of the first rhythm recovery episode, 0.58 (0.46, 0.8) ng/mL, was below the therapeutic interval. This confirms the almost complete transplacental transfer of sotalol and the significant limitation in the case of digoxin. Previously, ABCB1/P-glycoprotein had been shown to limit fetal exposure to drugs. Further studies (including multicenter ones) to clarify the genetic features of the transplacental pharmacokinetics of antiarrhythmic drugs are needed.

## 1. Introduction

Fetal disorders of the heart rhythm and conduction occur with a frequency from 0.1 to 5% from all pregnancies and may cause low cardiac output, fetal hydrops, intrauterine growth restriction, premature labor, and fetal, neonatal, and infant mortality [1,2]. The most common fetal arrhythmias (FA), >160–180 beats/min, include premature atrial contractions (PACs), the supraventricular tachycardia (SVT), and the atrial flutter (AFL). SVT and AFL are complicated by nonimmune fetal hydrops in 30–50% of FA pregnancies resulting in the perinatal mortality increase of up to 17–43% [3,4,5].

The timely initiation of ART promotes the conversion of fetal tachyarrhythmia to the sinus rhythm, the regression of the concomitant non-immune fetal hydrops, and the cardiac remodeling indicators, thus decreasing the rate of perinatal morbidity and mortality. Transplacental ART contributes to the prolongation of pregnancy to full term with an average delivery at 39.8 ± 1.2 weeks [6]. Transplacental medical interventions, including digoxin, flecainide, sotalol, and amiodarone, started more than 40 years ago. Still, there is no consensus on the first-line treatment of FA, especially SVT. Different medical centers present the significant variability in the evaluation of drug-induced rhythm recovery: from 1–5 days (M = 1 day) [7], 1–35 days (M = 7.5 days) [8], and up to 12 days [9]. There are significant differences in the frequency of fetal tachycardia conversion to sinus rhythm complication with non-immune fetal hydrops (38–86%) and without hydrops (27.8–96%) [3,7,8,9]. The need of optimal treatment regimen search for the fetus with tachyarrhythmia is of high importance.

Digoxin, cardiac glycoside, is often chosen as the first-line agent for the treatment of fetal SVT without fetal hydrops. It is relatively safe and has a long history of use [10]. The initial dosage of digoxin is 0.5–2.0 mg/day with subsequent correction [11,12]. Digoxin monotherapy showed a lower effective rate (27.8%) than combined with flecainide/sotalol (72.2%) [11]. The transplacental transfer of digoxin is reduced in fetal hydrops [13,14]. The drug bioavailability at absorption from the gastrointestinal tract is 70–80%. Digoxin is metabolized in the liver and excreted mainly by the kidneys (50–70% is unchanged) [12,15].

Sotalol, a non-selective β-blocker, is used both as monotherapy and in combination with digoxin/flecainide in fetal arrhythmia [5,7]. The initial dosage is 160–320 mg divided b.i.d., in the absence of medical cardioversion—up to 480 mg/day [7,11]. The oral bioavailability of sotalol is 89–100%. The drug is not metabolized by the liver and is excreted in 80–90% unchanged by the kidneys. According to Van der Heijden et al. the use of sotalol contributed to rhythm recovery in 78% of fetuses with SVT [6]. In the absence of a therapeutic effect within 5–8 days, the addition of digoxin or flecainide led to a successful conversion of the arrhythmia in 100% of cases [6,8].

Pharmacogenetic differences due to different levels of expression and activity of enzymes responsible for the acetylation, hydrolysis, oxidation, or metabolism of drugs shows a wide ethnic variation [16]. The patients can be divided in two groups: rapidly and slowly metabolizing drugs. The *ABCB1* (*MDR1*, multidrug-resistance gene) gene is located at the 7q21 locus and consists of 28 exons. The clinical significance was found for three *ABCB1* single nucleotide polymorphic (SNP) loci: 1236C > T (rs1128503), 3435C > T (rs1045642), and 2677G > T/A (rs2032582). These SNPs change the level of P-glycoprotein expression and the excretion of a number of drugs, in particular digoxin [16]. P-glycoprotein (Pgp), encoded by the *ABCB1* gene, is an ATP-dependent membrane transporter, exporting some drugs and xenobiotics from the inside of endothelial cells to the outside. Pgp protects the cells and organs against from toxic xenobiotic agents and some drugs. It is expressed in the liver, small and large intestines, kidneys, and placenta [17,18]. 

*ABCB1* 3435C > T polymorphism change the level of Pgp expression [19]. The patients with the 3435TT genotype have a low expression of the P-glycoprotein in the intestine, liver, kidneys, and placenta. Medium therapeutic doses of drugs result in higher concentrations of drugs in the blood due to an increase in bioavailability (more complete absorption by small and large intestine enterocytes, decrease in the barrier function of the placenta, and inhibition of excretion by the kidneys and liver) [20].

The frequency of 3435C > T polymorphism of the *ABCB1* gene in the Russian Federation is 30.0% (24.8–35.6) for the TT genotype, 48.6% (42.7–54.5) for the CT genotype, and 21.4% (16.8–26.6) for CC [21,22]. Every third patient has a risk to develop side effects; every fifth patient metabolizes drugs quickly and most likely does not receive the expected therapeutic effect. An individual approach (low therapeutic doses) for patients with the 3435TT *ABCB1* genotype is preferable. Digoxin, a first-line ART for fetal heart rhythm disorders, with a very narrow recommended therapeutic range, is a substrate for P-glycoprotein. *ABCB1* genotyping can individualize the choice of the digoxin dosing regimen. 

Digoxin and sotalol are drugs with a delayed onset of equilibrium concentration. With long-term use, a linear increase in the level of drugs is noted. The widely used and recommended therapeutic concentration range of digoxin for adults is very narrow (0.8–2 ng/mL) [23] and, recently, it was proposed to limit this range up to 0.5–0.9 ng/mL [24]. Controversial data on fetal/maternal serum digoxin ration were presented—from 40% up to 90% [11]; in particular, Takekazu Miyoshi et al. showed 0 0.53 (0–1.0) value [25] and, according to Ebara H. et al., it was 0.84 (0.6–1.0) [26]. The ratio of the fetus to maternal blood level for sotalol was found at 1.11 (0.67–2.87) by Oudijk M.A. et al. [27] and 1.07 (47.2–371.6) by Takekazu Miyoshi et al. [25]. The relatively high ratio of sotalol in the amniotic fluid to the fetus blood, 3.2 (1.28–5.8), could be explained by the active renal excretion of the drug, which is not metabolized by liver [27]. The digoxin ratio in the amniotic fluid to the fetus blood was found at 6.0 (4.0–8.0), according to Miyoshi [25].

The accurate and precise monitoring of the antiarithmic drugs concentrations (for instance, digoxin) remains of utmost importance. In general, immunoassays are routinely employed for digoxin monitoring in mother’s blood [28], but these techniques have relatively low specificity toward glycosides (i.e., digoxin) with high levels of cross-reactions with digoxin-like immunoreactive factors [26,29,30,31,32]. In modern laboratories, an increasing number of studies are carried out using the gold standard for diagnostics, namely high-performance liquid chromatography with tandem mass spectrometry (HPLC-MS/MS). There are several reports concerning the application of this analytical approach for the determination of digoxin in adults and newborns [33,34,35,36,37,38,39].

There are few HPLC-MS/MS studies of digoxin and sotalol pharmacokinetics for adults and neonates. The transplacental pharmacokinetics and pharmacodynamics of common tachyarrhythmia drugs differ significantly from those for adults due to the presence of three blood circulations (maternal, fetal, and placental) and the complexity of their interaction [26,27,40,41,42,43,44,45]. It is necessary to choose the personal dosing regimen to avoid the mother/fetal intoxication or the lack of the antiarrhythmic effect [46,47]. The cancellation of drugs for transplacental therapy may be due to the fetus state and the development of side effects in a pregnant woman. The adverse effects occurred with a frequency of 30–38.8% and included mild headache, moderate nausea, diarrhea, and may be reduced or eliminated by reducing the dose [23,48]. There is a genetic predisposition to the antiarrhythmic drugs variations in pharmacokinetics, pharmacodynamics, and side effects [20,21,40,49,50,51]. Of particular interest is the influence of the xenobiotic metabolization gene, *ABCB1*, polymorphisms to the transplacental ART efficacy. 

The aim of this study was to study the pharmacokinetics of transplacental antiarrhythmic drugs using HPLC-MS/MS, as well as to assess the effect of the *ABCB1* 3435C > T polymorphism on the efficacy and maternal/fetal complications of digoxin treatment.

## 2. Results

### 2.1. Comparative Clinical Data between Pregnant Persons and Controls

The main group, FA, included 89 patients; the control group included 50 healthy pregnant women. The average age was 31 ± 3 years and was not statistically different between the groups. These groups were comparable in anthropometric data, parity, and gynecological history. The pregnant of the control group were statistically less likely to have chronic diseases (*p* < 0.001) compared to the FA group. The risk of preterm delivery was significantly higher in the group with FA (*p* < 0.001). In the early neonatal period, in the groups of pregnant women with fetal/neonatal arrhythmia, a significantly longer hospitalization was required (13 (9 and 22) days) vs. 4 (3 and 5) days, *p* < 0.001). Table 1 summarized the clinical data of the groups studied.

### 2.2. Efficacy of Transplacental Fetal Treatment

The average time for fetal tachyarrhythmia onset was 30.4 ± 3.8 weeks. Supraventricular tachyarrhythmia was recorded in 55.1% of the FA cases, premature atrial contractions were diagnosed in 29.2% of fetuses with FA, and atrial flutter in 15.7% (Table 2). The non-immune fetal hydrops and ascites were more common in the SVT group (*p* < 0.05). Each pregnant woman with FA performed dynamic monitoring of fetal “heart rate” using fetal pocket Doppler. The episodes of rhythm recovery during ART were regarded as incomplete rhythm conversion (85.71–91.89% of the cases). The complete conversion to sinus rhythm was achieved in 34.6–71.4% of the FA cases at 6.3 ± 4.2 days. It was significantly more common in groups with SVT and AFL (*p* = 0.027). In the treatment of AFL, the effect of fetal antiarrhythmic therapy was considered positive with a decrease in the frequency of ventricular contractions to 120–150 beats/min, despite the persistence of atrial flutter.

### 2.3. The 3435C > T Polymorphism of the ABCB1 Gene in Pregnant Women and Newborns

The 3435C > T polymorphism of the *ABCB1* gene was screened in the blood of 30 pregnant women and 18 newborns to determine its influence at the course of pregnancy and fetal/antenatal tachyarrhythmia. The CC genotype was found in nine (30%) women and TT in eight (26.7%) women. Among 18 newborns, 4 (22.2%) carried the 3435CC variant and 3 (16.7%) the 3435TT *ABCB1* gene polymorphism.

Adverse maternal effects (dyspepsia, digestive symptoms, and headache) complicated the ART of seven pregnant women (23.3%). The incidence of side effects in pregnant women with the CC genotype was 3 times higher than the carriers of the T allele (44.4% vs. 14.3%, *p* = 0.056). The most pronounced decrease in the frequency of the maternal heart rate was found on the 12th day of fetal ART in T allele carriers (62 (58 and 68) vs. 70 (69 and 72), *p* = 0.011).

An earlier response to the transplacental therapy was noted in the group of fetuses with the T allele (6 (4 and 7) days) compared with the CC genotype (10 (4 and 14) days). The frequency of rhythm disruptions under digoxin medication differed significantly in fetuses with the homozygous 3435C > T genotype of the *ABCB1* gene (*p* = 0.033). Thus, in four (100%) fetuses with the CC genotype, there were no episodes of rhythm disruption. The fetuses with the 3435TT *ABCB1* variant disrupted the rhythm at the attempt of the digoxin dosage decrease in 100% of cases.

### 2.4. HPLC-MS/MS for Digoxin and Sotalol Measurement Optimization and Validation

Two separate sample preparations and HPLC-MS/MS procedures for sotalol and digoxin measurement were developed. These analytes differ significantly in chemical parameters (in particular, hydrophobicity), which required the use of different chromatographic columns and separation conditions. The concentration in clinical samples at ART for sotalol is three orders of magnitude higher than for digoxin. The simultaneous analysis of sotalol and digoxin would significantly reduce the sensitivity of the digoxin level measurement. The protein precipitation for sample preparation resulted in worse sample clean-up than liquid-liquid extraction. To obtain high specificity of sotalol and digoxin HPLC-MS/MS analysis in complex matrix (plasma, urine, and amniotic fluid), three daughter ion fragments for each precursor ion were chosen (Table 3). A typical chromatogram was obtained and is presented in Appendix A.

The blank sample (plasma and urine) HPLC-MS/MS analysis from 10 healthy volunteers showed the absence of any additional chromatographic peaks with the similar retention time of the analytes and internal standards. The HPLC-MS/MS methods proved to be linear (r > 0.99) within 0.2–10 ng/mL, 10–1000 ng/mL, 0.5–10 ng/mL for digoxin, and 0.2–10 µg/mL, 10–1000 µg/mL, and 0.2–10 µg/mL for sotalol in plasma, urine, and amniotic fluid, accordingly. The lower limits of quantitation (LLOQs) for digoxin in plasma, urine, and amniotic fluid were 0.2 ng/mL, 7.5 ng/mL, and 0.2 ng/mL, accordingly. The LLOQs for sotalol in plasma, urine, and amniotic fluid were 0.2 µg/mL, 10 µg/mL, and 0.2 µg/mL, accordingly. The accuracy and precision of digoxin and sotalol HPLC-MS/MS measurements did not exceed 13%. The matrix effect for digoxin ranged from 76% to 110% for digoxin in plasma and from 85% to 92% in urine. For sotalol, the matrix effect was 81–114% and 83–94% in plasma and urine, respectively. The extraction recovery variance of sotalol and digoxin did not exceed 9% (the median values were from 82% up to 90% for different matrixes). The validation results were summarized in Appendix A.

### 2.5. Digoxin Pharmacokinetics in ART

In this study, the drug monitoring during the ART with narrow therapeutic window drugs (digoxin and sotalol) was applied for an optimal dosing. The digoxin and sotalol concentrations were determined in the plasma blood, urine, and amniotic fluid of 30 pregnant women four times (from the beginning of ART, 3rd (2–5) day, up to delivery) and in the plasma cord blood of 30 newborns at delivery using HPLC-MS/MS-MRM methods (Table 2 and Table 3). All the data obtained, dosages, and dates were presented in Appendix A.

The progress of digoxin accumulation in the maternal blood, urine, and amniotic fluid was estimated (Figure 1). The level of digoxin in the blood and urine of the mother rose a little in the beginning of ART (from 3rd (2–5) days to the 9th (9, 10) day) and decreased at delivery. The level of digoxin in amniotic fluid increased at delivery compared to the second observation point (6th (4, 8) day after the start of ART). These differences were not significant (*p* > 0.05). Figure 2 illustrated the correlation of the digoxin level between different maternal fluids (blood, amniotic fluid, and urine of) and cord blood at delivery.

The highest level of correlation (r_S_ = 0.78) was observed between the digoxin concentration in maternal and cord blood at delivery. It was possible to calculate the drug concentration in the mother’s blood (r_S_ = 0.75) using the urine digoxin level. The correlation was relatively low between the digoxin concentration in the cord blood and in the amniotic fluid (Figure 2). The highest level of digoxin in the blood, urine, and amniotic fluid were achieved in the group of pregnant women with TT ABCB1 polymorphism at the first and second time points followed by a decrease to the third point and delivery, but the difference was statistically non-significant.

Figure 3 illustrates the digoxin concentration in the mother’s blood and cord blood at delivery regardless of the maternal/newborn genotype. The regression curve (r_S_ = 0.78) was calculated using the Passing Bablok linear regression method:

Digoxin_b_ = −0.04 + 0.44 × Digoxin_m_, where Digoxin_b_ was the digoxin concentration in cord blood and Digoxin_m_ was the digoxin concentration in maternal blood.

The observed linear relationship was used to calculate the digoxin concentration in the baby blood using the level in the mother’s blood. The drug concentration in the fetus blood at rhythm recovery (first observation point) was evaluated (Table 4). The level of the drug in the fetus blood at the first rhythm recovery episode was 0.58 (0.46 and 0.80) ng/mL, which is below recommended for adults’ therapeutic range for digoxin—0.8–2.0 ng/mL.

Figure 3 contained one outline point. The significant deviation of the predicted digoxin mother/fetus ratio, 1:0.44, was found in the CC genotype for both mothers and babies (1:2). A more accurate regression equation was calculated for the case of the TT/TC genotype of the child. The resulting model was described by the expression: Digoxin_b_ = −0.2 + 0.6 × Digoxin_m_, r_S_ = 0.87.

The 3435CC ABCB1 variant was found for four newborns under study. A dispersion analysis showed the absence of a statistically significant effect of the mother’s 3435C > T ABCB1 genotype on the digoxin level in the biological fluids (Appendix A). Table 5 presented the newborn genotype influence on the digoxin concentration for woman’s plasma, amniotic fluid, and cord blood at delivery. The digoxin concentration in the amniotic fluid at delivery was significantly lower (*p* < 0.05) in the 3435CC ABCB1 variant of the child. The digoxin level in the women’s blood and cord blood did not significantly differ with respect to the baby ABCB1 genotype (*p* > 0.05).

### 2.6. Sotalol Pharmacokinetics in ART

The evaluation of sotalol transplacental pharmacokinetics was performed using HPLC-MS/MS in the MRM mode. Figure 4 presented the drug concentration in pregnant women’s blood, urine, and amniotic fluid during the ART. The median sotalol concentration was approximately constant at the first three time points in the maternal blood and urine. The drug level in maternal blood and urine decreased at delivery due to the dosage reduction. The *ABCB1* gene did not affect the transplacental sotalol pharmacokinetics according to the available data. 

Figure 5 presented the results of the correlation analysis. The level of correlation between the sotalol in the woman’s blood and cord blood, in the amniotic fluid and maternal blood, and in the amniotic fluid and the cord blood was high at delivery time.

It was possible to calculate the fetal sotalol using the mother’s drug concentration in the blood (Figure 6) or amniotic fluid:
$${\mathrm{Sotalol}}_{b}\;=\;0.98\;\times \;{\mathrm{Sotalol}}_{m},$$
$${\mathrm{Sotalol}}_{a}\;=\;0.41\;+\;4.8\;\times \;{\mathrm{Sotalol}}_{b},$$
where sotalol_b_ was the concentration of sotalol in the cord blood, sotalol_m_—in the maternal blood, and sotalol_a_—in the amniotic fluid.

The sotalol ratio in the fetus blood to the maternal blood was 1.0 (0.97, 1.07). Thus, the level of sotalol in the mother’s blood was the same as in the fetus/newborn; the drug penetrated the placental barrier completely. In the first rhythm recovery episode, the concentration of sotalol in the fetus’s blood was 0.76 (0.49, 1.13) µg/mL, which is almost eight times higher than at the delivery. 

## 3. Discussion

The distribution of the fetal arrhythmia types and the average time for diagnosing (30.4 ± 3.8 weeks) in this study was consistent with the data available: PAC was presented in 29.2% and SVT and AFL in 70.2% of FAs [3,5,9,52,53,54,55,56,57]. A case series of viral disease during pregnancy and subsequent fetal heart rhythm disturbance described by Hoi-Shan Chan et al., 2001, Savarese et al., 2008, Takahashi et al., 2011, and Dejog et al., 2015 attracted attention to the infectious status of a pregnant woman [58,59,60,61,62,63]. More frequent acute respiratory viral infections (*p* = 0.025), asymptomatic bacteriuria (*p* = 0.053), and impaired vaginal microflora (*p* = 0.002) at the II trimester of pregnancy was found in the FA group compared to the control.

The complete medical rhythm conversion was achieved in 34.6–71.4% of cases and was significantly more often in SVT and AFL groups (*p* = 0.027), which is comparable to the data from other studies (27.8–96% and 38–86% in the absence and presence of non-immune fetal hydrops, accordingly) [3,5,7,9]. The frequency of rhythm disruption on drug dosage reduction was more frequent in the fetal SVT group (*p* = 0.015). It confirms the difficulties in individual selection of the initial transplacental ART and the drug dosage reduction regimen [3,27,64]. The side effects requiring discontinuation and/or reduction in the dosage were 1.6 times less common than published by Moatassim et al., 2018 and Chimenea et al., 2021 (23.3% vs. 30–38.8%) [23,48]. These side effects (dyspepsia, digestive symptoms, and headache) were eliminated with a decrease in the dosage of the drug.

The polymorphisms of genes coding the transporters involved in detoxification changes the individual features of the drug pharmacokinetics [17,18,21,50]. The data obtained confirmed the expediency of genotyping a pregnant woman for the 3435C > T polymorphism of the *ABCB1* xenobiotic detoxification gene. The predisposition to a decrease in the drug bioavailability in patients with a homozygous 3435CC variant showed a probable association with the development of ART side effects. Dynamic electrocardiography showed a pronounced decrease in “heart rate” in the women with the 3435TT allele of the *ABCB1* gene. This can be explained by the lower expression of the ABCB1/P-glycoprotein in the intestine, liver, kidneys, and placenta. When prescribing medium therapeutic doses of P-glycoprotein substrates (namely, digoxin), a higher concentration of the drug in blood is observed due to the more complete absorption by enterocytes of the small and large intestines and the inhibition of excretion by the kidneys and liver [17,18,19].

The expediency of fetal 3435C > T genotyping of the *ABCB1* gene in fetal arrhythmias had also been shown. The homozygous 3435TT variant in the fetus showed a probable association with an earlier response to ART and rhythm disruptions on the digoxin dosage reduction. The confirmation of the *ABCB1* 3435C > T polymorphism influence on the course of fetal arrhythmia requires a multicenter, prospective, randomized, and controlled trial.

The features of drug pharmacokinetics during pregnancy are due to physiological changes in the mother’s body and the presence of an additional fetoplacental circulation affecting the distribution, metabolism, and elimination of the drug [41,42,65]. The fetal invasive examination is limited. A non-invasive method to determine the antiarrhythmic drug concentration in the fetal blood based on pregnant women plasma or urine drug levels possesses a high value.

The enzyme-linked immunosorbent assay (ELISA) remains the most common method for digoxin measurement. This analytical method has several limitations, including the low specificity and insufficient limit of quantification. The high-performance liquid chromatography and tandem mass spectrometry (HPLC-MS/MS) is the “gold standard” for therapeutic drug monitoring. The advantages of the HPLC-MS/MS over the enzyme immunoassay method are high accuracy, sensitivity, specificity, and ability to measure drugs in various types of the biological samples (blood, urine, amniotic fluid, etc.). HPLC-MS/MS guarantees the optimal sensitivity to clarify the transplacental drugs pharmacokinetics [66] in the case of low concentrations (ng/mL for digoxin).

The HPLC–MS/MS method developed in this study was used to measure digoxin and sotalol blood plasma, amniotic fluid, and urine with adequate sensitivity and selectivity, high accuracy and precision, and a low matrix effect [38,66,67,68,69,70,71,72]. The high selectivity was obtained by the usage of three MRM transitions for each compound studied (analyte and IS). Similar to most of the previously published HPLC-MS/MS methods, digitoxin, *Digitalis* glycoside, was used as an internal standard for digoxin [38,66,67,68,69,70,71,72]. The digoxin LOQ was 0.2 ng/mL, which is lower than the widely used therapeutic range for adults (0.8–2 ng/mL). A low digoxin level, 0.42 (0.20, 0.66) ng/mL, was found in the cord blood of newborns with transplacental ART at delivery. 

The results obtained in the study of the pharmacokinetics of antiarrhythmic drugs showed a high degree of correlation between the level of digoxin and sotalol in maternal and cord blood. It allowed for fetal therapeutic drug monitoring using the mother plasma HPLC-MS/MS-MRM analysis. The “therapeutic window” achievement for digoxin in the blood of a pregnant woman expects a proportional level in the fetal blood plasma. The ratio of digoxin in cord blood to maternal blood, 0.35 (0.26:0.46), was lower than previously published by Takekazu Miyoshi et al. (0.53 (0 and 1.0)) [25] and Ebara H. et al. (0.84 (0.6 and 1.0)) [65]. These differences may be explained by drug withdrawal before delivery and the drug metabolism peculiarities under the multidrug-resistance gene polymorphisms. More accurate regression between fetus and maternal digoxin in blood was calculated after excluding babies with fast drug metabolisms (3435CC genotype of *ABCB1* gene). The fetus/maternal digoxin ratio for TC and TT babies was 0.38 (0.26 and 0.45) (r_S_ = 0.87).

The level of digoxin in the blood of the fetus at the first rhythm recovery episode was lower (0.58 (0.46 and 0.80) ng/mL) than the widely used and recommended therapeutic range for adults (0.8–2.0 ng/mL). This may be the reason for the need of the second antiarrhythmic drug (sotalol). 

The digoxin ratio in the amniotic fluid to the fetus blood was 3.37 (2.23 and 5.93) at the 8–10 day of ART and 10 (7.87 and 14.9) at delivery, which corresponds to the multicenter trial by Takekazu Miyoshi (6.0 (4.0–8.0)). The highest concentration of digoxin in the amniotic fluid was found in the fetal TC and TT genotype compared to the CC genotype (*p* = 0.031). These differences may be explained by the slower elimination of the antiarrhythmic drug in fetuses with the 3435CC *ABCB1* variant.

The sotalol ratio in fetus cord blood to maternal plasma was 1.0 (0.97:1.07), which is consistent with data from other studies: 1.11 (0.67–2.87) by Oudijk M.A. et al. [27] and 1.07 (47.2–371.6) by Takekazu Miyoshi et al. [25]. It should be expected that there are almost identical sotalol levels in the blood of mothers and the fetuses after achieving equilibrium. The ratio of the sotalol in amniotic fluid to fetal plasma was 4.88 (4.84 and 9.29) at delivery and 3.75 (1.68 and 5.77) at the 6th (4 and 8) day of ART, similar to Oudijk M.A.’s study at 3.2 (1.28–5.8) [27]. The pronounced accumulation of sotalol in the amniotic fluid may be the result of the effective renal excretion of the unchanged drug. This study confirmed the high efficacy of the placental transfer of sotalol (almost 100%) and the absence of individual patient characteristics’ (including the *ABCB1* genotype) effect on the pharmacokinetics of the drug. Thus, sotalol ART is more predictable for the mother and fetus. Several studies have shown the advantage of sotalol as a first-line drug in fetal antiarrhythmic therapy [13,27]. Sotalol, a non-selective β-blocker, prevents high blood pressure in pregnant women also [73].

The efficacy of the drug transplacental transfer depends on the placental perfusion, the pH values of the maternal and fetal blood, and the physicochemical properties of the drug (size, protein binding, hydrophobicity, etc.). The level of most drugs “behind the placenta” (fetal blood and amniotic fluid) varies from 20 to 80% of the level in the mother’s blood [26,27]. The immaturity of the fetal kidneys leads to the elimination of drugs by back diffusion through the placenta into the mother’s bloodstream. Part of the drug metabolites become more polar, impairing transplacental excretion, and accumulating in various tissues of the fetus. The maturation of the fetus urinary system increases the excretion of drugs into the amniotic fluid. The fetus swallows the drug and its active metabolites from the amniotic fluid, thus increasing the effect of the drug [26,27].

The pharmacogenetic differences in the xenobiotic detoxification gene (*ACBC1*) varies significantly [20,41,46,74]. Single nucleotide polymorphisms divide patients into rapidly and slowly metabolizing medicines [75]. An individual approach in the dosing of ABCB1/glycoprotein-P substrates is recommended, in particular, to reduce the dose of digoxin [46,76,77] for the 3435TT ABCB1 variant. The success of transplacental antiarrhythmic therapy largely depends on the drug dosage penetrating the placenta to the fetus. P-gp minimizes the transplacental transfer of substrates, enhancing the barrier function of the placenta [78]. The significant decrease in placental P-gp expression with advancing gestation results in the increase in the digoxin effect on the fetus in late gestation [17]. The down regulation or inhibition of P-gp expression in fetuses with 3435CC *ABCB1* polymorphism may improve the access of digoxin to the fetus and enhance the antiarrhythmic effect.

This study had several limitations. First, this was a single center prospective study. The main advantage of prospective data and sample collection was a single treatment protocol. However, the fetal ART was not randomized between several institutions. This was the largest prospective study in the limited number of cases per year (89 participants) [57]. However, therapeutic drug monitoring of digoxin/sotalol (HPLC-MS/MS) and ABCB1 gene polymorphism detection was performed for a limited number of patients (30 mother-child pairs). Thirdly, only 3435C > T polymorphism of the multidrug-resistance gene was studied in this work. Other polymorphisms of *ABCB1* and genes responsible for the acetylation, hydrolysis, oxidation, or metabolism of drugs may also influence the pharmacokinetics and pharmacodynamics of antiarrhythmic drugs. Further multicenter studies are needed to establish the most effective approaches to fetal ART.

## 4. Material and Methods

### 4.1. Study Design

The clinical part of the work was carried out by the 1st Department of Obstetric Pathology of Pregnancy of the National Medical Research Center for Obstetrics Gynecology and Perinatology named after the academician V.I. Kulakov of the Ministry of Healthcare of the Russian Federation. The samples were collected from October 2018 to January 2021. The study included 89 pregnant women with a prenatally diagnosed fetal heart rhythm disorder of the tachyarrhythmia type and ART and 50 healthy pregnant women (the control group). All the patients signed a voluntary informed consent to participate in the study. This work was approved by the Ethical Committee of the National Medical Research Center for Obstetrics, Gynecology and Perinatology named after Academician V.I. Kulakov (protocol No. 9, dated 22 November 2018).

The fetal arrhythmia was established on the basis of a change in the frequency and/or regularity of the fetal “heart rate” during auscultation or ultrasound examination and confirmed by echocardiographic examination. The criteria for inclusion were the fetal arrhythmias diagnoses established using ultrasound examination and confirmed with echocardiography, singleton pregnancy, informed agreement to participate in the study, and fetal antiarrhythmic drugs medication. The patients with multiple pregnancies, chromosomal abnormalities in the fetus, and maternal diseases excluding the use of the fetal therapy were not included. The exclusion criteria were the lack of desire/ability to continue the participation in the study as well as the serious side effects of ongoing therapy.

All the patients underwent laboratory (blood electrolytes, thyroid hormones, and glycated hemoglobin levels) and instrumental (electrocardiography, expert ultrasound examination of the fetus, Doppler ultrasonography of the vessels of the uteroplacental and fetal-placental blood flows, cardiotocography, and echocardiographic examination of the fetus) studies. In case of fetal tachyarrhythmia, a diagnostic transabdominal amniocentesis was performed under ultrasound-assistance in order to detect an infectious (viral and bacterial) etiological factor in the development of the pathology.

All the patients had different dosages of antiarrhythmic drugs during treatment (Appendix A). The starting dose of digoxin for all the patients was 0.75 mg/day and decreased with the development of side effects and/or according to the results of routine therapeutic monitoring of digoxin. The starting dose of sotalol was 160 mg/day and, in the absence of a therapeutic effect, increased to 480 mg/day. The daily dosages of drugs were provided in Appendix A for each sample collection time coinciding with the patient’s visit to the doctor. During the time interval between samplings (Table 6), the dosage did not change. The digoxin medication time was at 07.00/15.00/23.00 and sotalol at 08.00/20.00. The maximum concentration of digoxin/sotalol was achieved in 1–2 h and 2–3 h, respectively. For the simultaneous measurement of these drugs, the sampling was performed three hours after the last medication, at 10.00.

The HPLC- MS/MS-MRM studies of plasma blood, amniotic fluid, and cord plasma blood were performed four times during fetal ART for 30 pregnant women and newborns to assess the transplacental pharmacokinetics of sotalol and digoxin. Table 6 illustrated the type of samples at four time points collected for this group. The dates of sample collection were available in Appendix A. For these 30 pregnant women and 18 newborns, the 3435C > T polymorphism of the ABCB1 gene was screened also. The number of patients who underwent therapeutic drug monitoring decreased from 89 to 30 due to missing several visits to the doctor (the absence of samples at more than one time point), patient refusal of amniocentesis, or delivery in another medical center (especially in preterm birth). The first sample was collected at the 3rd (2 and 5) day, the second at 6th (4 and 8) days, and the third at the 9th (9 and 10) day.

The following HPLC-MS/MS investigations were performed for each pair of mother/fetus: 4 tests of the pregnancy blood, 4 tests of the pregnancy urine, 1 test of the newborn blood, and 2 tests of the amniotic fluid. The sum amounts of the samples were 99 maternal plasma blood, 97 maternal urine, 40 samples of the amniotic fluid, and 19 samples of the plasma cord blood.

### 4.2. Sample Collection

The amniotic fluids samples were obtained during transabdominal amniocentesis under ultrasound control and at delivery. All the surgical interventions were performed by one team of surgeons. The amniotic fluid was collected in 15 mL plastic tubes followed by centrifugation (10 min at 12,000 rpm at 40 °C). The supernatant was collected in 2 mL cryotubes (1.7–2 mL each). 

The maternal venous blood was extracted into a 3 mL EDTA tube followed by centrifugation at 3000 rpm in 20 min at 40 °C, the supernatant was centrifugated at room temperature in 10 min at 12,000 rpm and collected in 2 mL cryotubes (0.5–1 mL each). The newborn cord blood was extracted into a 1 mL EDTA tube. The sample preparation procedure was the same as the maternal blood. 

The average portion of urine (not the morning one) was collected under aseptic conditions in a plastic container (15 mL) followed by centrifugation (10 min at 12,000 rpm at 40 °C) and the supernatant were extracted into 2 mL cryotubes (1.7 mL each).

The processing and storage were carried out within 30 min after extracting the biomaterial. The labeled tubes were frozen at −80 °C and stored until the measurements.

### 4.3. Sample Preparation for HPLC-MS/MS

The measurements of the concentration of the antiarrhythmic drug (digoxin and sotalol) in the plasma of a pregnant woman and a newborn, urine, and amniotic fluid of a pregnant woman were performed by HPLC (Agilent, Santa Clara, CA, USA) with tandem mass-spectroscopy (SCIEX, Framingham, MA, USA). 

The MTBE (≥99.5%) HPLC grade was obtained from Fisher Chemical (Loughborough, UK). The methanol (MeOH) (99.9%) HPLC grade was obtained from Scharlab S.L. (Barcelona, Spain). The ultrapure deionized water was obtained with a Milli-q reference water purification system (Millipore Corporation, Billerica, MA, USA). The acetonitrile (99.9%) HPLC grade was obtained from Fisher Chemical (Loughborough, UK). The formic acid (98%), ammonium acetate (98%), sotalol hydrochloride analytical standard (308.82 g/mol), digoxin analytical standard (780.94 g/mol), digitoxin analytical standard (764.94 g/mol), and atenolol analytical standard (266.34 g/mol) were obtained from Merck KGaA (Darmstadt, Germany). All the stock solutions were prepared by dissolving the required amount of the soltalol/atenolol/digoxin/digitoxin in 100% MeOH. The solutions were stored in glass vials at −80 °C in a refrigerator prior to use. 

The sotalol sample preparation procedure was the following: 100 µL of a sample and 10 µL of the internal standard (atenolol) were mixed; the 1800 µL of ACN:MeOH = 3:1 mixture was added; and, after 5 min centrifugation, 1000 µL of the supernatant was transferred into a vial for HPLC-MS/MS analysis. The digoxin sample preparation procedure included the addition of 20 µL IS (digitoxin) to 400 µL; 1 mL MTBE addition to the mixture and centrifugation for 5 min; drying of 850 µl of the supernatant in the N2 stream at room temperature; and precipitate dilution in 100 mL of ACN:H_2_O = 1:1 mixture followed by centrifugation for 5 min with subsequent transfer to a vial with an insert for HPLC-MS/MS analysis. The plasma blood of the healthy volunteer was used as a matrix for the plasma blood HPLC-MS/MS analysis. The urine of the healthy volunteer was used as a matrix for the urine and amniotic fluid analysis.

### 4.4. HPLC-MS/MS Parameters

The chromatographic separation was carried out in an HPLC system Agilent 1260 infinity (Agilent, Santa Clara, CA, USA) consisting of a degasser, a pump, an autosampler, and the column with a thermostat. The mass spectrometer was calibrated according to standard ABSciex procedure using PPG Positive solution (2 × 10^−7^ M) from an ABSciex calibration kit. The solution was injected at 5 µL/min. The list of the monitored masses included *m*/*z* 59.050, 175.133, 500.380, 616.464, and 906.673. The tuning was performed until the mass error becomes less than 0.1 and peak width—between 0.6–0.8. The procedure was repeated for both quadrupole 1 and 3 at scan rates of 10, 200, 1000, and 2000 Da/s.

A Waters SPHERISORB column (Waters, Milford, Massachusetts, USA) 2.1 × 150 mm × 5 µm without a guard column was used for sotalol measurement. Ultrapure H_2_O with the ammonium acetate (10 mmol/L) was used as a mobile phase A. Undiluted acetonitrile (100%) was used as a mobile phase B. The measurements were performed in isocratic mode during 5 min at a constant phase ratio A/B 40/60. The flow was 800 µL/min. The injection volume was 5 µL. The column temperature was 40 °C. The system dead time was less than 30 s under these conditions and the retention time was 2.3 min and 2.7 min for sotalol and IS, respectively.

A Poroshel 120 EC-C18 column (Agilent, Santa Clara, CA, USA) 2.1 × 150 mm × 2.7 µm was used for the digoxin analysis with a guard column Zorbax (Agilent, Santa Clara, CA, USA) 2.1 × 5 mm × 1.9 µm. Ultrapure H_2_O with formic 0.1% acid and the ammonium acetate (10 mmol/L) was used as a mobile phase A. An ACN:H_2_O = 9:1 mixture with 0.1% formic acid and ammonium acetate (10 mmol/L) was used as a mobile phase B. The measurements were performed during 7 min: the first 1.5 min the ratio A/B was 9:1, after it initially increased to 5/95 and maintained a constant during 3 min then instantly returned to the initial parameters and maintained a constant during 2.5 min. The flow rate was 500 µL/min. The injection volume was 25 µL. The column temperature was 25 °C. The system dead time was less than 30 s and the retention time was 3.3 min and 3.6 min for digoxin and IS, respectively. Table 3 illustrated the multiple reaction monitoring (MRM)-selected transitions of sotalol, digoxin, and its internal standards.

### 4.5. HPLC-MS/MS MRM Methods Validation

The HPLC-MS/MS MRM methods were validated per the ICH M10 bioanalytical method validation guidelines [79], including sensitivity (LLOD and LLOQ), selectivity, linearity, intra- and interday precision, matrix effect, recovery, accuracy, and robustness. The blank matrixes of plasma blood, urine, and amniotic fluid experiments were created as a mixture of 10 heathy volunteers’ blood and urine.

The calibration curve for digoxin in amniotic fluid (Appendix A) was based on nine points in the range from 0.5–10 ng/mL with four levels of quality control (QC). The calibration levels were 0.5, 0.6, 1, 1.5, 2.5, 5, 6, 7.5, and 10 ng/mL. The QC levels were 0.5, 1.5, 5, and 7.5 ng/mL. The chromatograms for the blank and the samples are illustrated in Appendix A.

The calibration curve for digoxin in blood plasma (Appendix A) was based on nine points in the range from 0.2–10 ng/mL with four levels of QC. The calibration levels were 0.2, 0.5, 1, 1.5, 2.5, 5, 6, 7.5, and 10 ng/mL. The QC levels were 0.6, 1.8, 5, and 7.5 ng/mL. The chromatograms for the blank and the samples are illustrated in Appendix A.

The calibration curve for digoxin in urine (Appendix A) was based on seven points in the range from 10–1000 ng/mL with four levels of QC. The calibration levels were 10, 25, 50, 100, 200, 500, and 1000 ng/mL. The QC levels were 10, 25, 200, and 500 ng/mL. The chromatograms for the blank and samples are illustrated in Appendix A.

The calibration curve for sotalol in amniotic fluid (Appendix A) was based on nine points in the range from 0.2–10 ng/mL with four levels of QC. The calibration levels were 0.2, 0.4, 0.5, 0.6, 1, 2, 4, 8, and 10 ng/mL. The QC levels were 0.2, 0.6, 5, and 7.5 ng/mL. The chromatograms for the blank and the samples are illustrated in Appendix A.

The calibration curve for sotalol in blood (Appendix A) was based on nine points in the range from 0.2–10 µg/mL with four levels of QC. The QC levels were 0.2, 0.6, 5, and 7.5 µg/mL. The chromatograms for the blank and the samples are illustrated in Appendix A.

The calibration curve for sotalol in urine (Appendix A) was based on seven points in the range from 10 to 1000 µg/mL with four levels of QC. The QC levels were 10, 30, 500, and 750 µg/mL. The chromatograms for the blank and the samples are illustrated in Appendix A.

All the calibration curves were created using linear weighted (1/×2) regression analysis of the normalized peak areas (drug / IS area) with a correlation coefficient above 0.99.

To evaluate the selectivity of 10 healthy volunteers’ human plasma and urine, the spiked LLOQs and high QCs were tested. A low limit of detection (LLOD) was found as the digoxin or sotalol concentration at the S/N ratio of more than three. The lower limit of quantitation (LLOQ) was calculated as the lowest amount of analyte, quantitatively determined with acceptable (<20%) precision and accuracy. To study intra- and interday precision and accuracy five replicates of four QCs (LLOQ, low QC, medium QC, and high QC) for digoxin and sotalol were analyzed at three different days. The accuracy of the method was found as intra- and interday percent deviation (DEV) from the QC concentration nominal value (Appendix A). The intraday precision was described in terms of √ (within-day mean square)/GM × 100%, the interday precision √ (between-day mean square/n)/GM × 100%, where GM (grand mean) was the mean of the observed concentrations across run days using one-way analysis. 

The matrix effect (%) and extraction recovery (%) were evaluated three times using the blank sample (ultrapure H_2_O with LLOQ, medium and high QC digoxin/sotalol, and IS) and the blank matrix spiked before and after the extraction (Appendix A). To estimate the robustness, we slightly modulated the HPLC conditions, such as the flow rate, mobile phase (ammonium acetate in phase A), and column temperature at about 10% range. The validation data for each analyte and matrix were summarized in Appendix A.

### 4.6. ABCB1 Gene Polymorphism Study

The *ABCB1* gene single nucleotide polymorphism (SNP) study of 3435C > T locus was carried out for 30 pregnant women and 18 newborns using a polymerase chain reaction (PCR) with a consequent readout of melting curves using a modified kissing probes (adjacent probes) assay [80]. The venous blood cell fraction (erythrocytes and leukocytes) was aliquoted in cryotubes and stored at −80 °C. The DNA extraction was performed using a Prep-GS-Genetics kit (DNA-Technology JSC, Moscow, Russia) according to instructions. The DNA concentration was measured on a DNA minifluorimeter (Hoefer, USA). The typical values were in the range from 50 to 100 µg/mL. The genotyping was carried out using the original labeled oligonucleotides for the modified kissing probe assay (DNA-Technology JSC, Moscow, Russia). The DNA amplification, fluorescence, and melting temperature measurements were performed on the DT-96 Real-Time PCR Cycler (DNA-Technology JSC).

### 4.7. Statistical Analysis

The statistical data processing was performed in the RStudio (1.383 GNU) using our own scripts in the R language (4.1.1). The Shapiro–Wilk test was used to check the normality of the data distribution. Median values (Me) and quartiles (Q1, Q3) were used to describe the quantitative data without normal distribution. The qualitative data were presented as absolute values (%). The comparative analysis for qualitative data was performed using a Fisher’s exact test, χ^2^-test. The comparative analysis of the quantitative data was carried out using the Mann–Whitney test for the pairwise comparison of groups and the Kruskal–Wallis test for the comparison of more than two groups. The significance threshold was determined to be 0.05.

The correlation coefficients between two quantitative variables were estimated using Spearman’s rank correlation. The modeling using the Passing-Bablok linear regression method was applied to develop a predictive model for a quantitative variable from another quantitative variable.

## 5. Conclusions

In our work, longitudinal therapeutic transplacental drug (digoxin and sotalol) monitoring was performed during fetal ART using the HPLC-MS/MS methods developed. The effects of the 3435C > T polymorphism of the multidrug-resistance gene *ABCB1* on the efficacy of fetal ART, maternal and fetal complications, and features of transplacental pharmacokinetics of digoxin were studied for the first time. The ratio of digoxin in the fetus to maternal blood was 0.35 (0.27 and 0.46) and, in the case of sotalol, it was 1.0 (0.97 and 1.07). The level of digoxin in the blood of the fetus at the time of the partial conversion of fetal tachycardia to sinus rhythm, 0.58 (0.46 and 0.8) ng/mL, did not reach therapeutic concentrations (0.8–2.0 ng/mL). This confirms the data of other studies on the almost complete transplacental transfer of sotalol and its significant limitation in the case of digoxin. One of the explanations for this fact may be the pronounced expression of P-glycoprotein, encoded by the *ABCB1* gene, by syncythiotrophoblasts: this protein performs a barrier function in the placenta. Further studies to clarify the genetic features of the transplacental pharmacokinetics of antiarrhythmic drugs are needed.

## Figures and Tables

**Figure 1 ijms-24-01848-f001:**
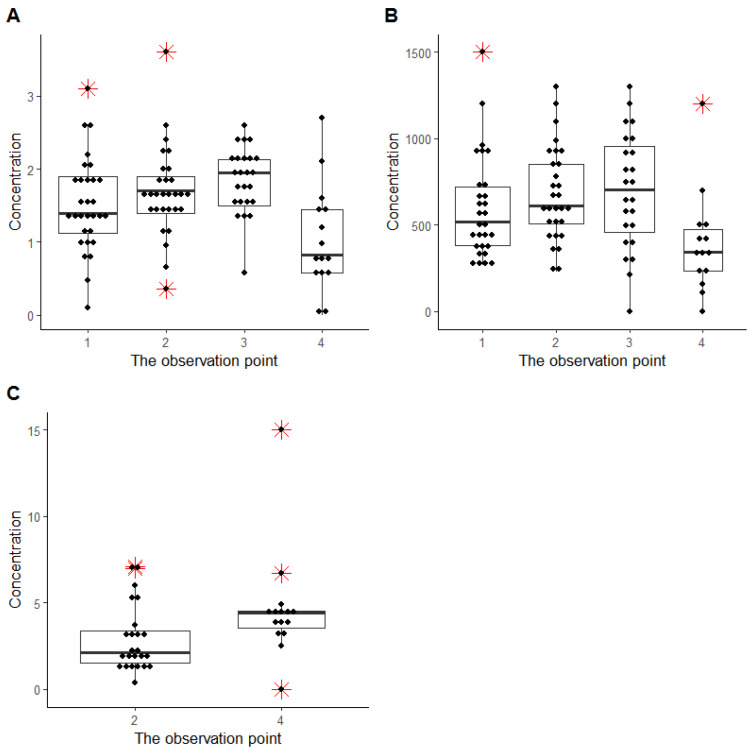
Digoxin concentration at the observation points, ng/mL: (**A**)—in the woman’s blood, (**B**)—in the woman’s urine, and (**C**)—in the amniotic fluid. The red symbol marked outliers.

**Figure 2 ijms-24-01848-f002:**
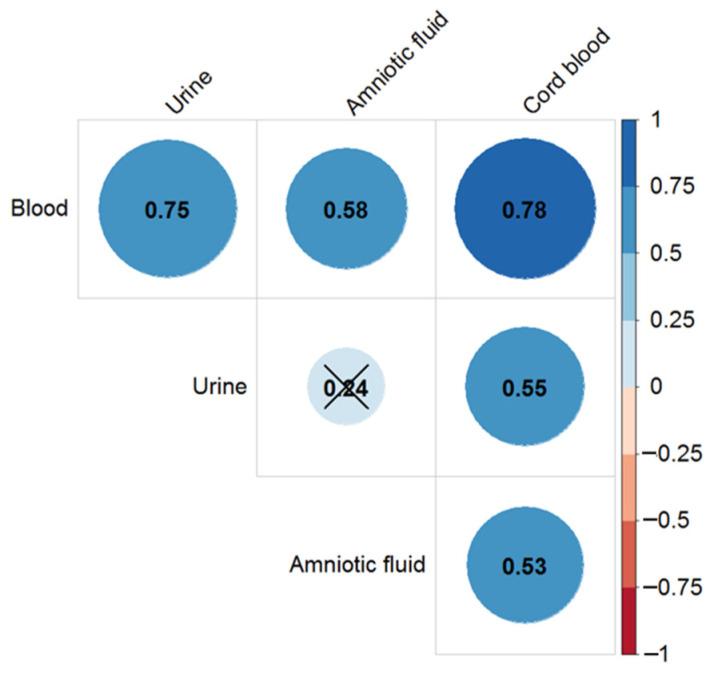
Correlation coefficients for the digoxin concentration in the blood, urine, and amniotic fluid of pregnant women and cord blood at delivery (fourth time point) with significance level *p* < 0.05. The cross (×) marked the insignificant correlation (*p* > 0.05).

**Figure 3 ijms-24-01848-f003:**
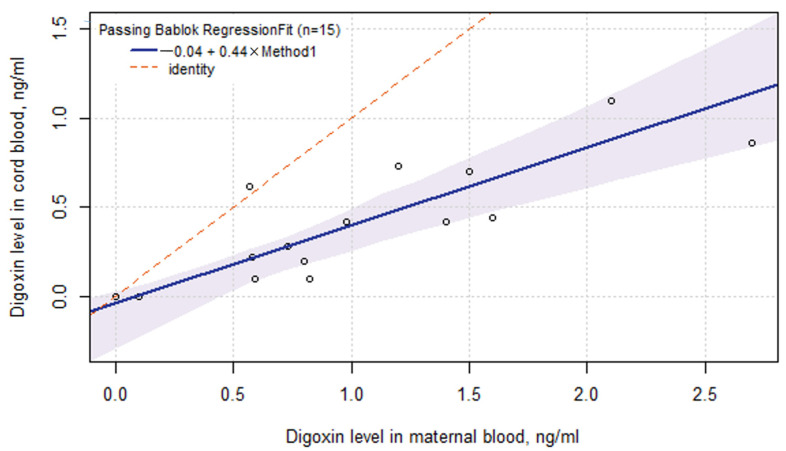
Linear regression between digoxin level in cord blood and maternal blood at delivery. The 0.95-confidence bounds were calculated using the bootstrap (quartile) method.

**Figure 4 ijms-24-01848-f004:**
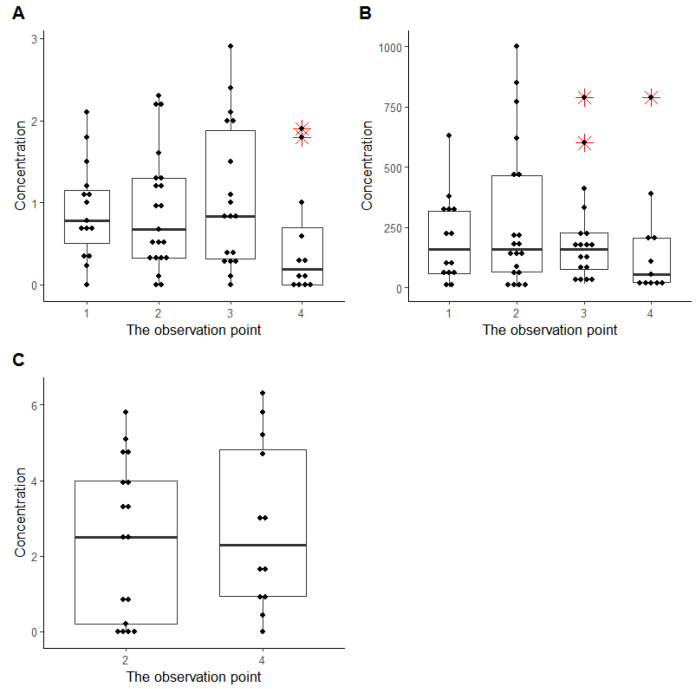
Sotalol concentration at the observation points, µg/mL: (**A**)—in the woman’s blood, (**B**)—in the woman’s urine, (**C**)—in the amniotic fluid. The red symbol marked outliers.

**Figure 5 ijms-24-01848-f005:**
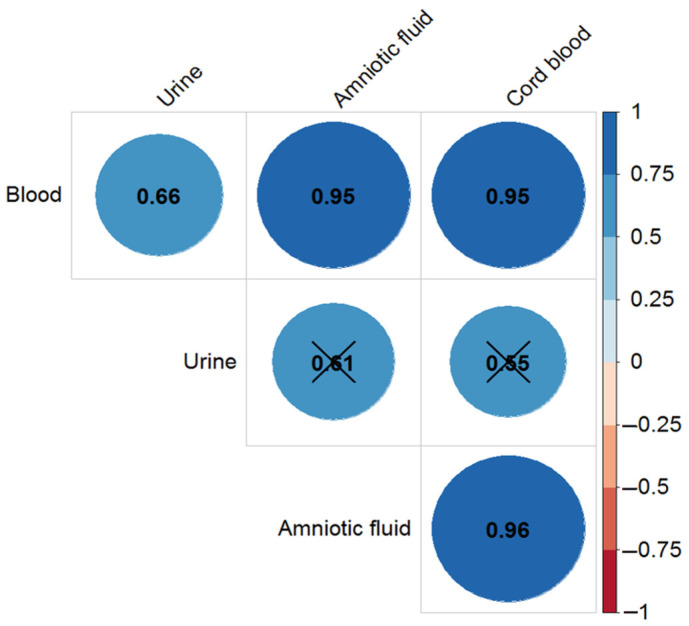
Correlation of sotalol level between different biological fluids at delivery. The cross (×) marked the insignificant correlation (*p* > 0.05).

**Figure 6 ijms-24-01848-f006:**
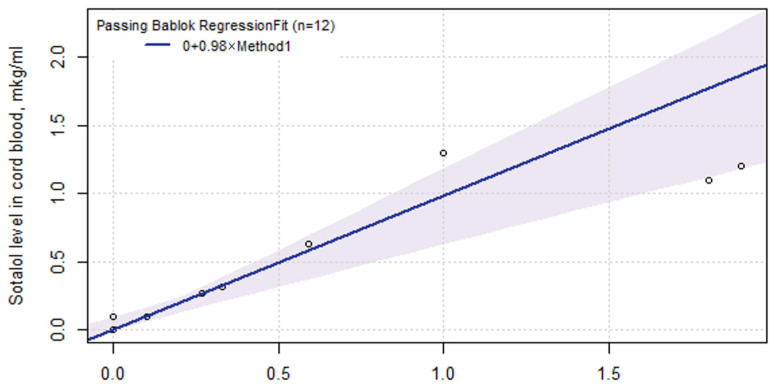
Linear regression between sotalol level in baby cord blood and maternal blood at delivery. The 0.95-confidence bounds were calculated using the bootstrap (quartile) method.

**Table 1 ijms-24-01848-t001:** Clinical data of pregnant women studied.

	Fetal Arrhythmia Group (n = 89)	Control Group (n = 50)	*p*-Value
Age (year)	31 (28–34)	31 (27–34)	0.31
Growth (m)	1.65 (1.62–1.68)	1.64 (1.60–1.69)	0.79
Weight (kg)	71 (64–80)	70 (64–77)	0.75
Pregnancy weight gain (kg)	11 (8–13)	12 (10–15)	0.19
Parity	2 (1–3)	2 (1–2)	0.99
Chronic diseases	77 (86.5)	25 (50)	<0.001
Toxicosis	19 (21.4)	10 (20)	0.85
Early pregnancy termination threat	12 (13.5)	8 (16)	0.69
Threat of preterm delivery in the third trimester	35 (39.3)	1 (2)	<0.001
Gestation age at delivery (weeks)	38^6^ (37^3^–39^2^)	39^0^ (38^4^–40^0^)	0.015
Newborn weight (g)	3240 (2860–3600)	3346 (3020–3565)	0.33

**Table 2 ijms-24-01848-t002:** Features of ART depending on the type of fetal arrhythmia. SVT—supraventricular tachycardia, PAC—premature atrial contraction, AFL—atrial flutter, FA—fetal arrhythmia, ART—antiarrhythmic therapy.

	SVT (n = 49, 55.1%)	PAC (n = 26, 29.2%)	AFL (n = 14, 15.7%)	*p*-Value
Sustained FA (n = 28)	19 (38.8)	0	9 (64.3)	<0.001
Nonimmune fetal hydrops (n = 10)	9 (18.4)	0	1 (7.1)	0.049
Ascites (n = 12)	11 (22.5)	0	1 (7.1)	0.018
Fetal ART (n = 64)	38 (77.6)	12 (47.2)	14 (100)	<0.001
Dose reduction in digoxin (n = 36)	25 (65.8)	3 (25)	8 (57.1)	0.046
Sotalol (n = 43)	32 (84.2)	3 (25)	8 (57.1)	<0.001
Dose reduction in sotalol (n = 30)	23 (60.5)	1 (8.3)	6 (42.9)	0.004
Rhythm recovery episodes (n = 57)	34 (91.9)	11 (91.7)	12 (85.7)	0.838
Complete rhythm recovery (n = 50)	31 (63.3)	9 (34.6)	10 (71.4)	0.027
Rhythm recovery time	5 (2–8)	2 (1–6)	6 (1–6)	0.486
Absence of rhythm disruption on ATR	23 (60.5)	10 (83.3)	9 (64.3)	0.304
Disruption of the rhythm on a drug dosage decrease	13 (34.2)	0	1 (7.14)	0.015
Gestation age at delivery	38.5 (37.32–39.1)	39.05 (38.42–39.62)	38.2 (37.62–39)	0.084

**Table 3 ijms-24-01848-t003:** Multiple reaction monitoring (MRM) selected transitions of digoxin, digitoxin, atenolol, and sotalol.

Q1, Da	Q3, Da	Time, ms	ID	DP, V	EP, V	CE, V	CXP, V
273.0	255.0	120.0	Sotalol1	41.0	10.0	17.0	22.0
273.0	213.000	120.0	Sotalol2	41.0	10.0	25.0	22.0
273.0	133.100	120.0	Sotalol3	41.0	10.0	35.0	14.0
267.1	145.1	120.0	Atenolol1(IS)	136.0	10.0	35.0	12.0
267.1	190.000	120.0	Atenolol2(IS)	136.0	10.0	25.0	16.0
267.1	74.100	120.0	Atenolol3(IS)	136.0	10.0	29.0	8.0
798.4	651.4	80.0	Digoxin1	41.0	10.0	19.0	12.0
798.4	781.5	80.0	Digoxin2	41.0	10.0	13.0	16.0
798.4	97.1	80.0	Digoxin3	41.0	10.0	51.0	8.0
782.3	97.1	80.0	Digytoxin1(IS)	76.0	10.0	33.0	12.0
782.3	635.5	80.0	Digytoxin2(IS)	76.0	10.0	17.0	20.0
782.3	243.1	80.0	Digytoxin3(IS)	76.0	10.0	21.0	22.0

**Table 4 ijms-24-01848-t004:** The concentration of digoxin (ng/mL) in the blood plasma of pregnant woman and the fetus at the rhythm recovery and at delivery. The concentration of digoxin in the fetus blood at the first rhythm recovery episode was estimated using the level in the plasma blood of the pregnant woman at the nearest observation point, ng/mL.

	Maternal Blood (ng/mL)	Fetus Blood (ng/mL)
First rhythm recovery episode	1.4 (1.1, 1.9)	0.58 (0.46, 0.80)
Delivery day	0.82 (0.58, 1.5)	0.42 (0.20, 0.66)

**Table 5 ijms-24-01848-t005:** The level of digoxin (ng/mL) in the blood, cord blood, and amniotic fluid of pregnant woman at delivery in accordance with child 3435C > T *ABCB1* genotype. * indicated that the differences between groups were statistically significant (*p* < 0.05) according to the Mann–Whitney criteria.

Newborn 3435C > T *ABCB1* Genotype	CC	TC and TT	*p*-Value
Pregnant women blood	0.34 (0.075, 1.1)	0.98 (0.76, 1.40)	0.1773
Cord blood	0.31 (0, 0.68)	0.42 (0.21, 0.57)	0.6468
Amniotic fluid	2.9 (1.9, 3.5)	4.5 (4.2, 4.8)	0.01076 *

**Table 6 ijms-24-01848-t006:** Time points and sample type for transplacental pharmacokinetics of sotalol and digoxin study.

	Pregnant Women Venous Blood	Pregnant Women Urine	Amniotic Fluid	Cord Blood
First observation point	+	+	-	-
Second observation point	+	+	+	-
Third observation point	+	+	-	-
Delivery	+ (with continued therapy)	+ (with continued therapy)	+	+

## Data Availability

Data are contained within the Appendix A.

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
