# Peer review of "Transplacental Therapeutic Drug Monitoring in Pregnant Women with Fetal Tachyarrhythmia Using HPLC-MS/MS"

_ijms, 2023, doi:10.3390/ijms24031848_

Round 1
Reviewer 1 Report
1) Lines 19-21: The sentence “Timely initiation of the transplacental antiarrhythmic therapy promotes to conversion of fetal tachycardia to sinus rhythm and the regression of the concomitant non-immune dropsy.” should be replaced with “Timely initiation of the transplacental antiarrhythmic therapy promotes conversion of fetal tachycardia to sinus rhythm and the regression of the concomitant non-immune dropsy.”
2) Line 58: Delete ‘to’ infront of conversion.
3) Line 85: Separate ‘thesubsequent’ to ‘the subsequent’.
4) Line 203: After Table 2, the next table indicated is Table 8. This is unacceptable since Tables should appear in an ascending order, i.e., from Table 2 to Table 3, etc. Please recheck and number the Tables appropriately.
5) Line 204: The reference Supplementary 1 contains figures and tables. It is therefore important for the authors to indicate the specific supplementary table or figure instead of just indicating Supplementary 1.
6) Line 219: There is no Table 3 in the manuscript. Authors should provide the Table 3.
7) Line 265: Change “(Table S1, Supplementary1)” to “(Table S1)”
8) Lines 322-323: Change “These side effects (dyspepsia, digestive symptoms and headache) eliminated with a decrease in the dosage of the drug.” to “These side effects (dyspepsia, digestive symptoms and headache) were eliminated with a decrease in the dosage of the drug.”
9) Line 431: Delete “named after Academician V.I. Kulakov”.
10) In Table 7: Change “Begginning” to “Beginning”
11) The authors should indicate how the mass spectrometer was calibrated to ensure accurate mass measurements.
12) It seems the symbol for micro is represented wrongly. Please check and insert the appropriate symbol.
13) The authors should provide information/data on how the HPLC-MS/MS MRM method was validated per ICH M10 bioanalytical method validation guidelines
14) The ethical approval number of the study should be indicated in the manuscript.
Author Response
1) Lines 19-21: The sentence “Timely initiation of the transplacental antiarrhythmic therapy promotes to conversion of fetal tachycardia to sinus rhythm and the regression of the concomitant non-immune dropsy.” should be replaced with “Timely initiation of the transplacental antiarrhythmic therapy promotes conversion of fetal tachycardia to sinus rhythm and the regression of the concomitant non-immune dropsy.”
Answer. Done.
2) Line 58: Delete ‘to’ in front of conversion.
Answer. Done.
3) Line 85: Separate ‘thesubsequent’ to ‘the subsequent’.
Answer. Done.
4) Line 203: After Table 2, the next table indicated is Table 8. This is unacceptable since Tables should appear in an ascending order, i.e., from Table 2 to Table 3, etc. Please recheck and number the Tables appropriately.
Answer. The table was moved to beginning and the renumeration was made.
5) Line 204: The reference Supplementary 1 contains figures and tables. It is therefore important for the authors to indicate the specific supplementary table or figure instead of just indicating Supplementary 1.
Answer. Done.
6) Line 219: There is no Table 3 in the manuscript. Authors should provide the Table 3.
Answer. Done.
7) Line 265: Change “(Table S1, Supplementary1)” to “(Table S1)”.
Answer. Done.
8) Lines 322-323: Change “These side effects (dyspepsia, digestive symptoms and headache) eliminated with a decrease in the dosage of the drug.” to “These side effects (dyspepsia, digestive symptoms and headache) were eliminated with a decrease in the dosage of the drug.”
Answer. Done.
9) Line 431: Delete “named after Academician V.I. Kulakov”.
Answer. It is the official title of the Center.
10) In Table 7: Change “Begginning” to “Beginning”.
Answer. Done.
11) The authors should indicate how the mass spectrometer was calibrated to ensure accurate mass measurements.
Answer. The following information about the mass spectrometer calibration was added to the Materials and Methods section: “The mass spectrometer was calibrated according to standard ABSciex procedure using PPG Positive solution (2*10-7 M) from ABSciex calibration kit. The solution was injected at 5 µl/min. The list of the monitored masses included m/z 59.050, 175.133, 500.380, 616.464, 906.673. The tuning was performed until mass error becomes less than 0.1 and peak width between 0.6-0.8. The procedure was repeated for both quadrupole 1 and 3 at scan rates 10, 200, 1000 and 2000 Da/s.”
12) It seems the symbol for micro is represented wrongly. Please check and insert the appropriate symbol.
Answer. Corrected.
13) The authors should provide information/data on how the HPLC-MS/MS MRM method was validated per ICH M10 bioanalytical method validation guidelines
Answer. The information was added in the Material and Methods section: “The HPLC-MS/MS MRM methods were validated per ICH M10 bioanalytical method validation guidelines, including sensitivity, selectivity, linearity, intra- and interday precision, matrix effect, recovery, accuracy (bias), and robustness.”, and “To evaluate selectivity 10 healthy volunteer human plasma and urine, spiked with LLOQs and high QCs were tested. Low limit of detection (LLOD) was found as digoxin or sotalol concentration at S/N ratio more than three. The lower limit of quantitation (LLOQ) was calculated as the lowest amount of analyte, quantitatively determined with accepta-ble (<20%) precision and accuracy. To study intra- and interday precision and accuracy five replicates of four QCs (LLOQ, low QC, medium QC and high QC) for digoxin and sotalol were analyzed at three different days. The accuracy of the method was found as intra- and interday percent deviation (DEV) from the QC concentration nominal value (Table S2, Supplementary1). The intraday precision was described in terms of √(within-day mean square)/GM*100%, the interday precision - √(between-day mean square/n)/GM*100%, where GM (grand mean) was the mean of the observed concentrations across run days by one-way analysis. Tables 2S-4S were included in the Supplementary 1”. Additional information was added in the Results section: “Blank samples (plasma and urine) HPLC-MS/MS analysis from 10 healthy volunteers showed the absence of any additional chromatographic peaks with the similar retention time of the analytes and internal standards. The HPLC-MS/MS methods proved to be linear (r>0.99) within 0.2-10 ng/mL, 10-1000 ng/mL, 0.5-10 ng/mL for digoxin and 0.2-10 µg/mL, 10-1000 µg/mL, 0.2-10 µg/mL for sotalol in plasma, urine and amniotic fluid, accordingly. The lower limits of quantitation (LLOQs) for digoxin in plasma, urine and amniotic fluid were 0.2 ng/mL, 7.5 ng/mL, 0.2 ng/ml, accordingly. The LLOQs for sotalol in plasma, urine and amniotic fluid were 0.2 µg/mL, 10 µg/mL, 0.2 µg/ml, accordingly. The accuracy and precision of digoxin and sotalol HPLC-MS/MS measurements did not exceed 13%. The matrix effect for digoxin ranged from 76% to 110% for digoxin in plasma and from 85% to 92% in urine. For sotalol the matrix effect was 81-114% and 83-94% in plasma and urine, respectively. The extraction recovery variance of sotalol and digoxin did not exceed 9% (median values were from 82% up to 90% for different matrixes). The validation results were summarized in the Table S4, Supplementary1.”.
14) The ethical approval number of the study should be indicated in the manuscript.
Answer. The necessary information was added at the of the article: “Institutional Review Board Statement: The study was approved by the Ethical Committee of the National Medical Research Center for Obstetrics, Gynecology and Perinatology named after Academician V.I. Kulakov (protocol No. 9, dated November 22, 2018).”
Reviewer 2 Report
My main criticism is that the methodology is not clear as regards how the doses were established and if the beginning of the treatment was intended as a rapid digoxin loading. I know that in Supplement 2 you have written the doses for each patient, but is that the loading dose or the daily dose (and for how many days)? I think this information should be in the article text.
Another concern is the moment for point 1. Please explain the difference between section 2.4 and table 7 concerning point 1 of sampling. Please explain when precisely the first sample was taken, you say “at the beginning of the transplacental ART” - that means after the first dose or for digoxin after 24 h, or after 6-8 days of drug intake. I see in Supplementary 2 that the interval between the first dose and the first sample point is larger than that you mentioned (6-8 days). For patient nr 5 is only 2 days, for patient 4 is 24 h, for patient 30 is 11 days. What doses were used during this time?
This has a major impact on the methodology if the therapeutic monitoring was intended to measure steady-state concentration.
Questions (Q) and observations (O):
Abstract:
¨ O1: Line 22 ‘Drugs resistance and detoxification genes are involved in the absorption and elimination of a large number of drugs’ is a phrase to general. It should be eliminated because the next phrase says the same thing Maybe use ”Polymorphisms of these genes coding the transporters involved in detoxification determines the individual features of the drug pharmacokinetics.”
¨ O2 (Line 29) ART – abbreviation not explained. You also used ”transplacental antiarrhythmic therapy” on line 36. Use only one (or the abbreviation or the full text). This observation is available for all text.
¨ O3 (line 40): It is mean concentration
Introduction:
¨ O4 (Line 55) Please use the the AFL abbreviation for atrial flutter and AF or AFib for atrial fibrilation. This observation is available for all text.
¨ Q1 (line 56-69): Is it a spelling error for ”hydrops” or it was meant to be dropsy? Dropsy as an old term for oedema is not appropriate, it is not a MeSH term.
¨ O5 (line 67): Please verify the reference [9], I think it is not the right one (I did not find the value that you presented, and it is a review article, not an original one)
¨ O6 (lines 72-73) Reference needed. Reference [11] mentioned a loading dose of: 1.5–2 mg over 24–48 h and maintenance: 0.375–1 mg/day. Also, other studies like https://doi.org/10.1080/17512433.2017.1344096 recommend an oral loading dose of 0.5 mg/8 h for 24 h followed by a maintenance regimen of 0.5 mg/12 h treatment. You cited only one loading dose, 0.75mg, the one that you used. Explain (in the methodology) why.
¨ O7 (line77-78) references are needed for the pharmacokinetic facts such as https://doi.org/10.1038/clpt.2008.1 or https://doi.org/10.1080/17512433.2017.1344096
¨ O8 (line 81-82) Other authors cited in your article [11] mentioned a loading dose for sotalol of 160–320 mg divided b.i.d.; Maintenance: increased to 480 mg/day. Again, you choose to present only one value, the one that you used but there are more, so please change and write the doses as an interval.
¨ O9 (line 107 ): Cite “ Every third patient slowly metabolizes drugs and develops side effects” - not clear for which drugs, which side effect. Maybe to reformulate "have a risk to develop side effects"
¨ O10 (line 124): according to Miyoshi
¨ O11 (line 143) “ moderate nausea, diarrhea, and eliminated with the…” replaced with ”and may be reduced or eliminated by reducing the dose”
¨ O12 (line 144) : “There is a genetic predisposition to the antiarrhythmic drugs pharmacokinetics and pharmacodynamics features” Phrase not clear: a predisposition to what? To side effects, to variation in PK or PD? Please reformulate.
¨ O13 (line 146) “xenobiotic metabolization gene”
Results
¨ O14 (line 152) You cannot call “Efficacy and side effects of transplacental foetal treatment “ if the comparison was between pregnant persons with foetal tachyarrhythmia and a control group. I suggest subchapters: 2.1 Comparative clinical data between pregnant persons and control 2.2 Efficacy and side effects of transplacental foetal treatment – you describe only the group of 89 with FA (beginning with line 162) 2.3 The 3435C>T polymorphism of the ABCB1 gene in pregnant women and new-borns and so on.... Please describe in the Methodology all the groups. (as I understand you have 2.1 89/50 FA/control; 2.2 89 FA (49/26/14) 2.3 30pregnant and 18 babies for genetic analysis)
¨ O15 (line 173) Table 2 Explain the abbreviation used in the table.
¨ O16 (Table 2) The p-value in the first line of the table has no value. The frequency should be in the top of the table beside the number of cases, like: SVT (n=49; 55,1%).
¨ O17 (Table 2) Here mention total cases of sustained FA, such as: “Sustained FA (n=28), SVT (n=19; 38.8%)” and so on for all variables in the table.
¨ O18 (Table 2) Please explain why you differentiated the 2 groups (oral and IV digoxin). I think they are not necessary. As I mentioned, the prescribed doses are missing in the text.
¨ Q2 (Table 2) You did not define “dose reduction in digoxin”. What value? After how many days? After the results of plasmatic concentration? It was a systematic a posteriori drug monitoring or it was only using clinical evaluation (adverse reactions)? This should be clarified in the methodology section.
¨ Q3 (line 177) Why did you not mention the heterozygote CT?
¨ O19 (line 181) Rare by the definition of adverse reaction is between 1 in 1,000 and 1 in 10,000 people. How did you define serious adverse reaction? Is it the definition used in pharmacovigilance or not? If not, please define.
¨ O20 (line 183) The most pronounced decrease in the frequency of maternal heart rate contractions . Pay attention to the double brackets used at heart rate and in other places like line 187. Please note that 62 is the mean
¨ O21 (lines 190-191) “ disrupted the rhythm with a dosage decrease in 100% of cases “ is not clear, please reformulate, I don’t understand what dosage decrease.
¨ O22 (line 207) Explain the abbreviation LLOQ (the lower limit of quantification), explain QC (line 211)
¨ O23 (line 223) please use the standard term in pharmacokinetics: steady-state concentration instead of „ a stationary level”
¨ O24 (line 266) „digoxin content concentration for woman’s plasma”
¨ O25 (line 413 ) replace „low therapeutic” with „to reduce the doses of ...”
Material and methods
¨ O26 Besides all the observations mentioned above, this section should also include data on how were chosen the drugs, what doses were used, if the digoxin fast-loading regimen was used. This aspect is not truly clear and is central to the methodology. It should be mentioned why, of the 89 pregnant women initially, only 30 were monitored for drug concentration.
¨ O27 (line 456) scheme of saturation = rapid loading ?
¨ O28 (line 457) Explain why you chose to measure drug concentration after 3 h because blood samples should be obtained at least 6 hours, but optimally 12 hours, after digoxin administration of digoxin to ensure complete distribution of the blood to the tissues. Serum digoxin concentrations measured before these times may be falsely elevated.
¨ O29 (line 526) This is the first time in the article that you mention atenolol and digitoxin. Explain why.
Author Response
My main criticism is that the methodology is not clear as regards how the doses were established and if the beginning of the treatment was intended as a rapid digoxin loading. I know that in Supplement 2 you have written the doses for each patient, but is that the loading dose or the daily dose (and for how many days)? I think this information should be in the article text.
Answer. Thank you for your valuable remarks concerning the treatment methodology. In the Materials and methods section the required information was added: “All patients had different dosages of antiarrhythmic drugs during treatment (Supplementary2). The starting dose of digoxin for all patients was 0.75 mg/day and decreased with the development of side effects and/or according to the results of routine therapeutic monitoring of digoxin. The starting dose of sotalol was 160 mg/day and, in the absence of a therapeutic effect, increased to 480 mg/day. Daily dosages of drugs were given in Supplementary2 for each sample collection time coinciding with the patient's visit to the doctor. During the time interval between samplings (Table 6), the dosage did not change.”
Another concern is the moment for point 1. Please explain the difference between section 2.4 and table 7 concerning point 1 of sampling. Please explain when precisely the first sample was taken, you say “at the beginning of the transplacental ART” - that means after the first dose or for digoxin after 24 h, or after 6-8 days of drug intake. I see in Supplementary 2 that the interval between the first dose and the first sample point is larger than that you mentioned (6-8 days). For patient nr 5 is only 2 days, for patient 4 is 24 h, for patient 30 is 11 days. What doses were used during this time? This has a major impact on the methodology if the therapeutic monitoring was intended to measure steady-state concentration.
Answer. Thank you for this notion. Indeed, the sampling time was not correctly estimated for the first time point. Appropriate corrections have been made to the text of the article, Table 6. The information about sample collection time points was included: “The dates of sample collection were available in the Supplementary2. … The first sample was collected at 3 (2, 5) days, the second – at 6 (4, 8) days and the third – 9 (9, 10) days.”
Questions (Q) and observations (O):
Abstract:
O1: Line 22 ‘Drugs resistance and detoxification genes are involved in the absorption and elimination of a large number of drugs’ is a phrase to general. It should be eliminated because the next phrase says the same thing Maybe use ”Polymorphisms of these genes coding the transporters involved in detoxification determines the individual features of the drug pharmacokinetics.”
Answer. The sentence was corrected.
O2 (Line 29) ART – abbreviation not explained. You also used ”transplacental antiarrhythmic therapy” on line 36. Use only one (or the abbreviation or the full text). This observation is available for all text.
Answer. Corrected across the manuscript.
+¨ O3 (line 40): It is mean concentration.
Answer. As mentioned in the Materials and methods section: “Median values (Me) and quartiles (Q1, Q3) were used to describe the quantitative data without normal distribution.”.
Introduction:
O4 (Line 55) Please use the AFL abbreviation for atrial flutter and AF or AFib for atrial fibrilation. This observation is available for all text.
Answer. The abbreviations were changed in the text.
Q1 (line 56-69): Is it a spelling error for ”hydrops” or it was meant to be dropsy? Dropsy as an old term for oedema is not appropriate, it is not a MeSH term.
Answer. Dropsy was replaced to fetal hydrops.
O5 (line 67): Please verify the reference [9], I think it is not the right one (I did not find the value that you presented, and it is a review article, not an original one).
Answer. The reference [9] was changed to Strizek B, Berg C, Gottschalk I, Herberg U, Geipel A, Gembruch U. High-dose flecainide is the most effective treatment of fetal supraventricular tachycardia. Heart Rhythm. 2016;13:1283–1288. doi: 10.1016/j.hrthm.2016.01.029.
???¨ O6 (lines 72-73) Reference needed. Reference [11] mentioned a loading dose of: 1.5–2 mg over 24–48 h and maintenance: 0.375–1 mg/day. Also, other studies like https://doi.org/10.1080/17512433.2017.1344096 recommend an oral loading dose of 0.5 mg/8 h for 24 h followed by a maintenance regimen of 0.5 mg/12 h treatment. You cited only one loading dose, 0.75mg, the one that you used. Explain (in the methodology) why.
Answer. The text was corrected, the reference https://doi.org/10.1080/17512433.2017.1344096 was added.
¨ O7 (line77-78) references are needed for the pharmacokinetic facts such as https://doi.org/10.1038/clpt.2008.1 or https://doi.org/10.1080/17512433.2017.1344096
Answer. The references were added.
¨ O8 (line 81-82) Other authors cited in your article [11] mentioned a loading dose for sotalol of 160–320 mg divided b.i.d.; Maintenance: increased to 480 mg/day. Again, you choose to present only one value, the one that you used but there are more, so please change and write the doses as an interval.
Answer. The text was corrected: “The initial dosage is 160–320 mg divided b.i.d., in the absence of medical cardioversion – up to 480 mg/day”.
O9 (line 107 ): Cite “ Every third patient slowly metabolizes drugs and develops side effects” - not clear for which drugs, which side effect. Maybe to reformulate "have a risk to develop side effects"
Answer. The sentence was reformulated in accordance to the recommendation.
O10 (line 124): according to Miyoshi
Answer. Corrected.
O11 (line 143) “ moderate nausea, diarrhea, and eliminated with the…” replaced with ”and may be reduced or eliminated by reducing the dose”
Answer. The text was change from “eliminated with the dosage decrease” to “may be reduced or eliminated by reducing the dose”.
¨ O12 (line 144) : “There is a genetic predisposition to the antiarrhythmic drugs pharmacokinetics and pharmacodynamics features” Phrase not clear: a predisposition to what? To side effects, to variation in PK or PD? Please reformulate.
Answer. The text was reformulated: “There is a genetic predisposition to the antiarrhythmic drugs variations in pharmacokinetics, pharmacodynamics and side effects”.
O13 (line 146) “xenobiotic metabolization gene”
Answer. Corrected.
Results
¨ O14 (line 152) You cannot call “Efficacy and side effects of transplacental foetal treatment “ if the comparison was between pregnant persons with foetal tachyarrhythmia and a control group. I suggest subchapters: 2.1 Comparative clinical data between pregnant persons and control 2.2 Efficacy and side effects of transplacental foetal treatment – you describe only the group of 89 with FA (beginning with line 162) 2.3 The 3435C>T polymorphism of the ABCB1 gene in pregnant women and new-borns and so on.... Please describe in the Methodology all the groups. (as I understand you have 2.1 89/50 FA/control; 2.2 89 FA (49/26/14) 2.3 30pregnant and 18 babies for genetic analysis)
Answer. An additional subchapter was included: “2.1 Comparative clinical data between pregnant persons and controls”. In the Materials and Methods section the description of clinical groups was corrected: “The samples were collected from October 2018 to January 2021. The study included 89 pregnant women with a prenatally diagnosed fetal heart rhythm disorder of the tachyarrhythmia type and ART, and 50 healthy pregnant women (the control group).”.
¨ O15 (line 173) Table 2 Explain the abbreviation used in the table.
Answer. The abbreviations were explained.
¨ O16 (Table 2) The p-value in the first line of the table has no value. The frequency should be in the top of the table beside the number of cases, like: SVT (n=49; 55,1%).
Answer. Corrected.
¨ O17 (Table 2) Here mention total cases of sustained FA, such as: “Sustained FA (n=28), SVT (n=19; 38.8%)” and so on for all variables in the table.
Answer. Corrected.
¨ O18 (Table 2) Please explain why you differentiated the 2 groups (oral and IV digoxin). I think they are not necessary. As I mentioned, the prescribed doses are missing in the text.
Answer. As recommended the differentiation into two groups (oral and IV digoxin) was removed from the Table 2.
¨ Q2 (Table 2) You did not define “dose reduction in digoxin”. What value? After how many days? After the results of plasmatic concentration? It was a systematic a posteriori drug monitoring or it was only using clinical evaluation (adverse reactions)? This should be clarified in the methodology section.
Answer. In the Materials and methods section the required information was added: “The starting dose of digoxin for all patients was 0.75 mg/day and decreased with the development of side effects and/or according to the results of routine therapeutic monitoring of digoxin.”
¨+ Q3 (line 177) Why did you not mention the heterozygote CT?
Answer. The heterozygote type CT was grouped with TT genotype in accordance with the analytical results.
¨ O19 (line 181) Rare by the definition of adverse reaction is between 1 in 1,000 and 1 in 10,000 people. How did you define serious adverse reaction? Is it the definition used in pharmacovigilance or not? If not, please define.
Answer. The text was corrected: “Adverse maternal effects (dyspepsia, digestive symptoms and headache) were rare - 23.3% (7 pregnant women).”.
¨ O20 (line 183) The most pronounced decrease in the frequency of maternal heart rate contractions . Pay attention to the double brackets used at heart rate and in other places like line 187. Please note that 62 is the mean
Answer. The double brackets were deleted. As mentioned in the Materials and methods section: “Median values (Me) and quartiles (Q1, Q3) were used to describe the quantitative data without normal distribution.”.
¨ O21 (lines 190-191) “ disrupted the rhythm with a dosage decrease in 100% of cases “ is not clear, please reformulate, I don’t understand what dosage decrease.
Answer. The text was clarified: “The fetuses with the 3435TT ABCB1 variant disrupted the rhythm at the attempt of digoxin dosage decrease in 100% of cases.”
O22 (line 207) Explain the abbreviation LLOQ (the lower limit of quantification), explain QC (line 211)
Answer. The explanations were added.
O23 (line 223) please use the standard term in pharmacokinetics: steady-state concentration instead of „ a stationary level”
Answer. This text was rewritten.
O24 (line 266) „digoxin content concentration for woman’s plasma”
Answer. Corrected.
O25 (line 413 ) replace „low therapeutic” with „to reduce the doses of ...”
Answer. Corrected.
Material and methods
¨ O26 Besides all the observations mentioned above, this section should also include data on how were chosen the drugs, what doses were used, if the digoxin fast-loading regimen was used. This aspect is not truly clear and is central to the methodology.
Answer. In the Materials and methods section the required information was added: “All patients had different dosages of antiarrhythmic drugs during treatment (Sup-plementary2). The starting dose of digoxin for all patients was 0.75 mg/day and decreased with the development of side effects and/or according to the results of routine therapeutic monitoring of digoxin. The starting dose of sotalol was 160 mg/day and, in the absence of a therapeutic effect, increased to 480 mg/day. Daily dosages of drugs were given in Supple-mentary2 for each sample collection time coinciding with the patient's visit to the doctor. During the time interval between samplings (Table 6), the dosage did not change. Digoxin medication time was at 07.00/ 15.00/ 23.00 and sotalol - at 08.00/ 20.00.”
It should be mentioned why, of the 89 pregnant women initially, only 30 were monitored for drug concentration.
Answer. The explanation was added: “The number of patients who underwent therapeutic drug monitoring decreased from 89 to 30 due to missing several visits to the doctor (the absence of samples at more than one time point), patient refusal of amniocentesis or delivery in another medical Center (especially in preterm birth).”
O27 (line 456) scheme of saturation = rapid loading ?
Answer. The text was removed.
O28 (line 457) Explain why you chose to measure drug concentration after 3 h because blood samples should be obtained at least 6 hours, but optimally 12 hours, after digoxin administration of digoxin to ensure complete distribution of the blood to the tissues. Serum digoxin concentrations measured before these times may be falsely elevated.
Answer. The following text was changed: “Digoxin medication time was at 07.00/ 15.00/ 23.00 and sotalol - at 08.00/ 20.00. The maximum concentration of digoxin/sotalol is achieved in 1-2 hours, 2-3 hours, respectively. For simultaneous measurement of these drugs, the sampling as performed three hours after the last medication – at 10.00.”.
O29 (line 526) This is the first time in the article that you mention atenolol and digitoxin. Explain why
Answer. Digitoxin and atenolol were used as an internal standards for digoxin and sotalol, respectively. The information was added in the text.
Round 2
Reviewer 2 Report
Line 204 Again, a rare adverse reaction by the definition is one at 1/10000-100.000 patients (Very common ADR occurs in more than 1/10 patients who take the drug. Common (frequent) > = 1/100 and < 1/10. Uncommon (infrequent) >= 1/1000 and < 1/100 and so on) See these articles: https://www.acpjournals.org/doi/abs/10.7326/0003-4819-140-10-200405180-00009 https://www.sciencedirect.com/science/article/abs/pii/S0140673600027999
23,3% is not rare. So, please change the phrase. It will be useful to classify the adverse reactions by severity (for example EMA definition for the serious adverse reaction is an adverse reaction that results in death, is life-threatening, requires hospitalisation or prolongation of existing hospitalisation, results in persistent or significant disability or incapacity, or is a birth defect).
Line 777: ” did not reach therapeutic concentrations (0.8-2.0 ng/ml)” You should reformulate because the digoxin therapeutic range for adults with heart failure is not universally established at these values (see the article that compares the range recommendations for adults (https://www.ncbi.nlm.nih.gov/pmc/articles/PMC3646412/ ) and is definitely not set for fetal antiarrhythmic treatment at this value.
Please add the limitations of this study
Author Response
Line 204 Again, a rare adverse reaction by the definition is one at 1/10000-100.000 patients (Very common ADR occurs in more than 1/10 patients who take the drug. Common (frequent) > = 1/100 and < 1/10. Uncommon (infrequent) >= 1/1000 and < 1/100 and so on) See these articles: https://www.acpjournals.org/doi/abs/10.7326/0003-4819-140-10-200405180-00009 https://www.sciencedirect.com/science/article/abs/pii/S0140673600027999 23,3% is not rare. So, please change the phrase. It will be useful to classify the adverse reactions by severity (for example EMA definition for the serious adverse reaction is an adverse reaction that results in death, is life-threatening, requires hospitalisation or prolongation of existing hospitalisation, results in persistent or significant disability or incapacity, or is a birth defect).
Answer. The phrase was changed: “Adverse maternal effects (dyspepsia, digestive symptoms and headache) complicated the ART of 7 pregnant women (23.3%).”.
Line 777: ” did not reach therapeutic concentrations (0.8-2.0 ng/ml)” You should reformulate because the digoxin therapeutic range for adults with heart failure is not universally established at these values (see the article that compares the range recommendations for adults (https://www.ncbi.nlm.nih.gov/pmc/articles/PMC3646412/ ) and is definitely not set for fetal antiarrhythmic treatment at this value.
Answer. The sentence was reformulated as recommended: “The level of digoxin in the blood of the fetus at the first rhythm recovery episode was lower (0.58 (0.46, 0.80) ng/ml) than the widely used and recommended therapeutic range for adults (0.8-2.0 ng/ml).”.
Please add the limitations of this study.
Answer. The limitations were added: “This study had a number of limitations. First, this was a single center prospective study. The main advantage of a prospective data and sample collection was a single treatment protocol. However, fetal ART was not randomized between several institutions. This was the largest prospective study in the limited number of the cases per year (n=89) [57]. However, therapeutic drug monitoring of digoxin/sotalol (HPLC-MS/MS) and ABCB1 gene polymorphism detection was performed for a limited number of patients (30 moth-er-child pairs). Thirdly, only 3435C>T polymorphism of the multidrug-resistance gene was studied in this work. Other polymorphisms of ABCB1 and genes responsible for acetylation, hydrolysis, oxidation or metabolism of drugs may also influence the pharmacokinetics and pharmacodynamics of antiarrhythmic drugs. Further multicenter studies are needed to establish the most effective approaches to fetal ART.”.